# Knowledge, Understanding and Satisfaction with the Implementation of the Performance Management System at a District Hospital in the Madibeng Subdistrict, South Africa

**DOI:** 10.3390/ijerph192114461

**Published:** 2022-11-04

**Authors:** Gontse Thobejane, Hendry van der Heever, Mathildah Mokgatle

**Affiliations:** Department of Health Systems Management, School of Public Health, Sefako Makgatho Health Sciences University, Tshwane 0208, South Africa

**Keywords:** performance, management, knowledge, understanding, satisfaction

## Abstract

The Performance Management and Development System (PMDS) is an essential asset in aligning the strategic objectives of both the National and Provincial departments of health with the individual targets and Key Responsibility Areas (KRAs) of the individual employees working in these departments. The system is not without challenges. The sources of contention can be with the rewards mechanism of the system, bias in the application of the system and the incorrect use of the system for punitive measures. The objective of this study was to determine the perceived knowledge, understanding and satisfaction levels of employees at Brits District Hospital regarding the PMDS. Another objective was to provide an intervention targeting the knowledge and understanding of the key concepts of the system. A quantitative cross-sectional and longitudinal action research method was applied by tracking the responses of 64 healthcare workers three times, firstly in a questionnaire and then in an intervention checklist. The respondents were selected via proportionate stratified sampling. Data were analysed via STATA statistical software package 13.0. The respondents had a poor knowledge and understanding of how to score the PMDS. The respondents where highly dissatisfied with the development of their weaknesses and the recognition of their strengths. Training needs to be prioritized on both the PMDS and the employees’ areas of weakness affecting their performance toward their assigned Key Responsibility Areas. Employees need to be recognised for areas in which they show strength.

## 1. Introduction

There are various models and theories of performance management, namely, the balanced scorecard model, the management by objectives model, the flow model, the three Es model, the expectancy-based motivational model, the performance contract model, the South African excellence model and the European foundation for quality management model [1,2,3,4]. In addition to these models are the goal-setting theory, the social cognitive theory, the reinforcement theory and the systems theory [1,2,3,4]. There is a dearth of literature which explicitly specifies which model the PMDS is aligned with. The PMDS is a performance management system applied in the government departments and institutions of the Republic of South Africa [5]. The system is designed to objectively manage employee performance, allow employee involvement in setting standards, develop employee skills, correct noneffective employee performance and reward very effective performance by employees [6]. The PMDS exists in order to link the individual employee’s performance and developmental requirements with the organisational targets [5]. This is conducted by ensuring that the individual employee’s workplan is aligned with the public service institution’s Annual Operational and Performance Plans, which are in turn based on the government’s Medium-Term Strategic Framework (MTSF) for that administrative period [5]. The Medium-Term Strategic Framework is a set of targets and plans coordinated to realise the goals of the ultimate strategic document of the South African Government, which is the National Development Plan 2030 [7]. For there to be a dispute, there needs to be dissatisfaction. When employees disagree and reject the outcome of the performance management process, this means that the process is not meeting the needs of managers and this, for one, leaves employees dissatisfied and with poor commitment, which affects organisational performance [8]. Time is spent on resolving disputes related to the PMDS, which means that time for actual work is lost. These disputes can go on for hours and still not be resolved and require the intervention of a more senior manager. Since the PMDS policy stipulates that a dispute between a supervisor and a subordinate must be escalated to the next-level manager if a resolution cannot be obtained, it is possible to have the whole line function resolving a PMDS dispute [9]. There are employees in the public health service who do not understand why they are not receiving a performance bonus [10]. Many employees perceive their managers as lacking the required amount of skill to discuss their performance and coach them [11]. Inadequately trained managers and managers who are too afraid to disappoint employees are some of the management-linked reasons for why performance management fails [8].

A review of 13 similar research studies in health care settings was conducted in the course of this study. The studies conducted in South Africa generally found that the PMDS was poorly implemented. There was a general theme of there being a poor understanding of the system, a lack of knowledge and a dissatisfaction with the monetary rewards system, and a culture of rating bias and of managers using the system as a punitive measure [5,8,11,12,13,14,15,16,17]. The studies conducted internationally touched wide-ranging issues such as motivation, satisfaction, training and support. It was observed that the participants showed awareness of the performance management systems and at least knew that they existed in their organisations. Very few of the studies included an intervention or action element, and most were descriptive in their nature. Furthermore, the studies were cross-sectional or focused on past data contained in documents and published articles. None of the studies were longitudinal or assessed the impact of an intervention over time [18,19,20,21,22,23]. 

The authors of [17] state that a performance management system is an ideal asset created to add organisational value by motivating and measuring the effectiveness and relevance of employee performance to the healthcare institution’s targets. Performance management systems enhance the entire performance of health systems by emphasising rational decision-making [22]. However, a poorly implemented performance management system can have a negative impact on employee morale and on overall employee satisfaction levels. This can cause a high rate of employee turnover [16]. Employees consistently rate performance management systems very low in employee satisfaction surveys [13]. According to [8], among the reasons for employees being dissatisfied with the performance management process are that managers are not adequately trained, that there is a lack of consistency between ratings, that the appraisers or managers wish to be liked by those they manage, that imprecise goals are set and that leaders fail to lead by example. 

On 31 May 2018, staff at Charlotte Maxeke Johannesburg Academic Hospital (CMJAH) carried out industrial action and made it difficult to govern the hospital [10]. Hospital employees, including nurses, clerks, porters, cleaners and general assistants, blocked the entrance roads, threw rubbish out of the bins inside the hospital and generally disrupted the hospital operations [10]. One worker stated that he had not received a performance bonus even though he had been working at CMJAH for five years. He was quoted as saying “They do not want to pay [performance bonuses]. We stretch ourselves every day and work overtime, but we are not rewarded for it”. He went on to say “We just want to send a message. Physically, no patients will be affected but we want to send a message to management” [10]. 

Challenges with the PMDS are not restricted only to Charlotte Maxeke Johannesburg Academic Hospital or just the Gauteng Province [8]. Before the industrial action at Charlotte Maxeke Johannesburg Academic Hospital took place, the media and bodies such as parliament had long been querying the effectiveness of the PMDS in delivering on its mandate of improved service delivery [8]. The North West Province had also had challenges with the PMDS. In his 2019/2020 North West health budget vote, the Member of the Executive Council (MEC) for Health Honourable Madoda Sambatha noted that an analysis of the PMDS situation had been completed and that the PMDS review team appointed by the Minister of Health in 2018 had completed its work and produced a report with findings and recommendations that the North West Department of Health (NWDoH) could implement [24].

The inability to meet the developmental needs of employees is one of the recurring reasons for low satisfaction levels among employees regarding performance management [25]. This research aims to investigate the Brits District Hospital (BDH) health workers’ understanding of and satisfaction levels regarding the PMDS. The study also aims to positively impact the PMDS process by highlighting and implementing appropriate PMDS principles to employees through devising a checklist based on the North West Provincial Government (NWPG) PMDS policy and principles. 

The study setting was Brits District Hospital. The hospital is a district and level 1 hospital providing basic clinical services to the community of the Madibeng municipality in the North West Province of South Africa. With an estimate of 500 employees, the hospital has a district and level 1 hospital human resource organisational structure spearheaded by managers with the Hospital Chief Executive Officer at the top.

Empirical research findings regarding the understanding, satisfaction and recommendations of the Brits District Hospital staff on the PMDS could assist in improving the implementation and review of the PMDS at Brits District Hospital. This could contribute to protocol and policy amendments and inputs. This research study further aims to answer questions on the understanding, satisfaction levels and recommendations of the Brits District Hospital staff regarding the PMDS. Furthermore, this study can help to implement a checklist based on North West Provincial Government PMDS policy and, lastly, to improve the understanding and the satisfaction of Brits District Hospital staff regarding the PMDS. 

## 2. Materials and Methods

Guided by the prevalence of quantitative cross-sectional designs using questionnaires as the method of data collection, the researcher deployed a quantitative cross-sectional, longitudinal and action design utilising a questionnaire and intervention checklist for data collection [19,21,26]. Part one of this study was a cross-sectional descriptive study conducted through a self-administered questionnaire (Appendix A). The questionnaire was administered to respondents during three separate rounds of performance reviews at Brits District Hospital in January 2020, April 2020 and August 2020, meaning that each respondent answered the questionnaire on three separate occasions. The questions aimed to gather information such as the respondents’ age, rank, department, profession, years of experience and years at Brits District Hospital. The questionnaire also captured more detailed information regarding the respondent’s understanding of the aims of the PMDS, his/her satisfaction with it, the challenges s/he faced with it and his/her recommendations as to how s/he would like the PMDS to be administered. The questionnaire was informed by the principles outlined in the North West Provincial Government PMDS policy. 

The purpose of this part of the study was to collect data on the respondents’ perceptions of their own, their managers’ and their subordinates’ understanding of the PMDS. Another purpose of this part of the study was to collect data on the respondents’ satisfaction levels with and recommendations for the PMDS at Brits District Hospital. The null hypothesis was that there is no significant relationship between the respondents’ perception of their understanding of the PMDS and their involvement in disputes. The alternative hypothesis was that there is a significant relationship between the respondents’ perception of their understanding of the PMDS and their involvement in disputes.

Part two of the study was an action research design in which an improvement intervention in the form of the North West Provincial Government PMDS policy-based checklist developed by the researcher was completed by the respondents during three separate rounds of the performance reviews at Brits District Hospital. The purpose of this part of the study was to make respondents aware of the theoretical and technical aspects of the PMDS policy and to apply these aspects during performance reviews. This was conducted in each round after the respondents had completed the questionnaire. 

The intervention checklist and the questionnaire were adjusted for the second review round to correct minor grammatical and formatting errors that became evident during the first-round review. Finally, the questionnaire was readministered for the second and third round of reviews to assess if there were any changes (positive or negative) in comparison to the first-round review after the use of the intervention checklist. The recollection and the analysis of data collected via the second and third administration of the questionnaires meant that the study then became a longitudinal study because the respondents were followed over time with periodic reassessment to determine the outcomes of an administered intervention [27]. 

The members of the population invited to participate in the study were hospital employees from the areas of labour (unions), administration (finance, HR, logistics, procurement) and clinic (nursing, medical, allied, pharmacy) on managerial, supervisory and production levels.

Stratified sampling was used. The participants were divided into strata firstly according to their profession and secondly according to their rank. The division was based on similarity, because participants of the same profession (e.g., nurses) were in one subgroup and/or participants of similar rank (e.g., middle managers) were in a subgroup together. A further determinant of eligibility was that participants should previously have undergone a North West Provincial Government PMDS review. Proportionate sampling was used, whereby the sample size of each stratum was aimed to be proportionate to the population size of the stratum [28]. The population size for this study was 506 employees. The total sample size for this population at a confidence level of 95% and a margin of error of 5 was 217 [29]. A total of 230 employees were approached to participate in the study, and 134 agreed to participate in the study. In total, 64 of the respondents completed the questionnaire in the first round, and of those 64 respondents, 41 completed the questionnaire in the second round, and of those 41, 31 completed the questionnaire in the third round. 

After the raw data had been captured through the administration of the questionnaire and the intervention checklist, they were entered into the Excel (Microsoft office 2010 version) spreadsheet (Appendix A) and cleaned. The data were then exported to the STATA statistical software package 13.0 for analysis [30]. Descriptive and inferential statistics were performed. Percentages and proportions were calculated for categorical data such as profession, rank and highest qualification. The mean measures of dispersion were determined for discrete data. One-way tables of association were utilised and *p* < 0.05 was considered significant for statistical association. The two-way tables of association were used to determine the significance of the association between a lack of understanding of the PMDS and the occurrence of disputes in the performance management process. 

## 3. Results

Data were collected over three rounds in order to track the development of the respondents’ responses post administration of the intervention checklist after every round of data collection. 

### 3.1. Purpose of Contracting

Throughout three rounds of data collection, a majority of the respondents incrementally (89%, 95% and 97%, respectively) and correctly selected the accurate description of the purpose of contracting. This indicated that the respondents knew the purpose of contracting.

### 3.2. Definition of Key Responsibility Areas

The majority of the respondents incrementally (72%, 84% and 97%) and correctly selected the accurate description of the Key Responsibility Areas throughout the three rounds of data collection (Figure 1). This indicated that the respondents knew the Key Responsibility Areas.

### 3.3. Purpose of the Midyear Review

There was also a percentage increase (47–67%) in the understanding of the midyear reviews throughout the three rounds. By the end of the study, the respondents displayed an improved understanding of the midyear reviews compared to the beginning of the study.

### 3.4. Knowledge of the PMDS Scoring

The results indicated that throughout all three rounds of data collection, the majority of the respondents (81%, 86% and 73%, respectively) were unable to score the PMDS (Figure 2). This means that the respondents could not accurately appraise their performance in accordance with how PMDS scores are meant to be calculated.

### 3.5. Satisfaction with the Final Review Process

Initially, 40% of the respondents selected that they had a low level of satisfaction with the final review process. This number reduced to 30% and 32%, respectively, in the second and third rounds of data collection. By the end of the study, similar percentages (32%, 32% and 29%) of respondents had indicated that they had low, medium and high levels, respectively, of satisfaction with the final review process. At the conclusion of the study, a lesser percentage of the respondents were dissatisfied with the final review process.

### 3.6. Satisfaction with the Level of Development of Staff Weaknesses

Incremental results were obtained in the first (62%), second (65%) and third (66%) rounds of data collection, respectively, indicating that the majority of the respondents selected that they that had a low level of satisfaction with the development of staff weaknesses at Brits District Hospital. 

### 3.7. Satisfaction with the Level of Recognition of Staff Strengths at Brits District Hospital

Results from the data collected over three rounds indicated an incremental trend of close to half (46%, 51% and 55%) of the respondents selecting that they had a low level of satisfaction with the recognition of staff strengths at Brits District Hospital. Respondents who selected that they had a medium level of satisfaction with the recognition of staff strengths composed the majority of the remainder of the responses from the respondents, with 38% in the first round, 43% in the second round and 34% in the third round of data collection. By the end of the study, over half of the respondents were dissatisfied with the recognition of staff strengths.

### 3.8. Personal Level of Understanding of the PMDS

Throughout all three rounds of data collection, the majority (50%, 62% and 52%, respectively) of the respondents selected that they perceived themselves to have a medium level of understanding of the PMDS. By the end of the study, the majority of the respondents perceived themselves to have an average understanding of the PMDS (Figure 3).

### 3.9. Subordinate’s Level of Understanding of the PMDS

During the first two rounds of data collection, the majority (44% and 43%, respectively) of the respondents selected that they perceived their subordinate to have a medium level of understanding of the PMDS. In the third round of data collection, the largest proportion (34%) of the respondents selected that subordinate’s had a low level of understanding of the system (Figure 4).

### 3.10. Manager’s/Supervisors’ Level of Understanding of the PMDS

The majority of respondents initially selected that their supervisors/managers had a medium level of understanding of the PMDS, which was recorded in the first and second rounds of data collection. However, data collected from the third round indicated that above half (52%) of the respondents viewed their supervisors/managers to have a good understanding of the system. As the study progressed, managers were perceived to have an improved understanding of the PMDS (Figure 5).

### 3.11. Two-Sample t-Tests on the Relationship between the Respondents’ Perception of Their Supervisors’ Understanding of the PMDS and Their Involvement in PMDS-Related Disputes

A statistically insignificant relationship (t (34) = 1.57, *p* = 0.12) was found between the group means determined through the two-sample t test. The combined mean (M = 2.04, SD = 0.73) of the respondents’ perception of their supervisor’s level of understanding of the PMDS and the respondents’ involvement in disputes were both within their respective confidence intervals. The means indicated that the majority of the respondents perceived their supervisors’ understanding of the PMDS to be on the medium level, and that the majority of the respondents had never been involved in a PMDS-related dispute. However, this relationship was found to be insignificant, as indicated by the *p*-value (0.12) after calculation of the t-statistic (t (34) = 1.57). This indicated that respondents in the third round of the study perceiving their supervisors to have a medium understanding were insignificantly less likely to be involved in disputes. Due to the *p*-value (0.12) indicating insignificance, the null hypothesis that there is no significant relationship between the respondents’ perception of their supervisors’ understanding of the PMDS and the respondents’ involvement in disputes is accepted. The alternative hypothesis that there is a significant relationship between the respondents’ perception of their supervisors’ understanding of the PMDS and the respondents’ involvement in disputes is rejected (Table 1).

## 4. Discussion

The majority of respondents were able to show an understanding of the purpose of contracting and what the Key Responsibility Areas are. Respondents were able to show their understanding of the PMDS in their majority, and this percentage increased as the rounds progressed and as the intervention checklist was administered. It is, however, important to note that a majority of the respondents were unable to correctly score the PMDS despite the administration of the intervention checklist which provided an explanation of the system’s scoring. 

Throughout the study, half of the respondents were not confident enough to state that they had a good understanding of the PMDS. This is worrying, given that respondents need to understand the PMDS well in order to use the system correctly and to its full potential. Furthermore, ref. [8] found similar results in a study in Pelonomi Hospital in the Free State, where 60% of the respondents were satisfied with their knowledge and understanding of their roles pertaining to PMDS. 

Of even more concern is that a majority of the managers or supervisors indicated that they viewed their subordinates to have a poor understanding of the PMDS. This is further compounded by the fact that by the end of the study, only close to half of the respondents viewed their supervisors to have a good understanding of the system. This is despite the fact that the supervisor is the one who is supposed to lead the performance management process between the supervisor and supervisee. This can affect the trust that the staff members have in the ability of their superiors to lead the PMDS process correctly. Furthermore, ref. [25] found similar results in their study when professional nurses reported that their managers lacked the skill to discuss their performance and coach them. Additionally, ref. [11] further stated that this creates a negative relationship between the employees and the managers and negatively affects performance management. However, by the end of the third round, 52% of the respondents viewed their superiors as having a good understanding of the PMDS. This is encouraging, as the intervention checklist aimed to target correct PMDS practices such as being observant that timelines are adhered to, that individual goals are aligned with departmental goals and that the PMDS is conducted in a consultative manner with consistent feedback. 

Throughout rounds one to three of data collection, the majority of the respondents either had low or medium satisfaction levels with the PMDS process, which included the contracting, midyear and final review phases. In addition, ref. [8] found similar results in a study conducted at Pelonomi Hospital in the Free State, where 60% of the respondents stated that they were dissatisfied with the PMDS assessment method. After administration of the intervention checklist, higher satisfaction levels with the contracting, midyear and final review phases in the third and final round of data collection were observed.

The majority of respondents remained consistent in selecting that they had low levels of satisfaction regarding the way their weaknesses were developed. This may have been caused by the lack of training budget for the staff in the department, which also leads to the staff not receiving any training on the PMDS. Moreover, ref. [25] also holds this view when stating that development interventions should include training health care managers and supervisors on how to conduct performance management interviews. Additionally, ref. [13] adds that training managers on how to conduct the performance management process also paves the way to holding accountable managers who do not comply with the principles of the performance management process. 

Half of the respondents incrementally indicated low satisfaction with the way their strengths were being recognised at Brits District Hospital. Currently, highly effective and effective performance is recognised by a 1.5% increase in salary and a performance bonus, although the performance bonus has since been discontinued [31]. It is clear that the respondents are not satisfied with the recognition they receive, as only an average of 6% of the respondents over three rounds of data collection were highly satisfied with the recognition they received. A study in Cambodia also found low satisfaction levels amongst health care workers due to the poor linking of individual performance with the rewards given to the individual [32]. Furthermore, 60% of respondents were also dissatisfied with the alignment of performance and rewards in a study conducted at Pelonomi Hospital in the Free State [8].

It is the view of the researcher that the PMDS is a management-by-objectives model. This model states that the manager and the employee must meet and agree on the objectives of the employee for the performance-assessment period [14]. During this study, the respondents were asked to indicate the purpose of contracting during the implementation of the PMDS. The majority of the respondents (89% in the first round, 95% in the second round and 97% in the third round) correctly indicated that the purpose of contracting was to develop a performance agreement between the supervisor and subordinate. The North West Provincial Administration PMDS policy of 2018 in subsection (k) of the Performance Contract description section [6] states that “The manager and the employee are required to take joint responsibility for the development of the employees’ Performance Agreement and Workplan.”

Further evidence that supports that the PMDS is a management-by-objectives model is indicated by the North West Provincial Administration PMDS policy of 2018. Subsection (h) of the Performance Contract description section [6] states that “the Departmental Strategic Plan, Departmental Service Delivery Improvement Plan, the Component’s Operational Plan and the employee’s Job Description must inform the development of the individual’s Performance Agreement.” This is a characteristic of the management-by-objectives model, as indicated by [8] when stating that the strength of the management-by-objectives model is its focus on translating the strategic objectives of the organisation into the operational goals of the specific organisational departments and individual employees.

The management-by-objectives model also further states that the goals agreed upon by the manager and the employee should be periodically reviewed to assess the advancement being made [14]. The PMDS is in line with this model, as indicated by the North West Provincial Administration PMDS policy of 2018 in subsection (a) of the Performance Review and Assessment section [6], which states that “one to one communication sessions must take place from time to time between supervisors and employees about the progress toward the achievement of the objectives agreed upon.” During this study, the respondents were asked to indicate the purpose of the midyear review during the implementation of the PMDS. An increasing number of the respondents (43% in the first round, 63% in the second round and 67% in the third round) correctly indicated that the purpose of the midyear review was for the supervisor and official to jointly review the officials’ performance, to reach agreement, attach ratings and to agree on the steps to be taken to improve performance.

## 5. Conclusions

In total, 45% of the respondents in the first round and 55% in the third round indicated that they had never received training on the PMDS. Whilst the intervention checklist used in this study cannot constitute adequate training, it is noteworthy that the administration of the checklist coincided with improved outcomes amongst the respondents. This suggests that training sessions should be organised in an effort to improve the PMDS outcomes. It is the view of the researcher that the in-service training should include refresher training courses offered on an annual basis to accommodate updates in policy and keep staff informed on how the PMDS should be administered. 

Some of the areas of weakness identified in this study were a cause for concern, such as the scoring of the PMDS. More should be conducted to incentivise and make achievable the fulfilment of the requirements for rewards when very effective performance has been achieved. These ideas regarding training and the recognition and reward of highly effective performance are basic concepts in the PMDS that need to be acted upon urgently in order to improve the PMDS at Brits District Hospital. Increasing workers’ knowledge through educational programmes at the inception, during the implementation and after the conclusion of the PMDS process are required as part of the PMDS process [12]. 

This study contributes to the discourse regarding how the implementation of the PMDS can be improved to be more effective. It does this by firstly providing research results indicating the deficiency of training opportunities on the PMDS, secondly by illustrating the dissatisfaction of staff by the unavailability of training opportunities and lastly by showing that an intervention checklist can improve the staff’s knowledge of the PMDS. Additionally, this study was able to illustrate that the PMDS is aligned with the management by the objectives model of performance management.

### 5.1. Limitations

One of the challenges of the study was for the respondents to respond to the questionnaire and intervention checklist three times, months apart. This challenge resulted in some respondents not responding during the second and third rounds of data collection. Only 64 respondents participated in this study conducted at one district hospital in the Madibeng Subdistrict of the North West Province of South Africa. This led to a further limitation of this study, namely, that the study was only conducted at Brits District Hospital, and therefore, the results cannot be generalised to the rest of the North West Province’s public hospitals or public hospitals in the rest of the Republic of South Africa. Further studies covering more hospitals across more districts can yield more representative results. 

This study was a quantitative study which aimed to ascertain and track the respondents’ knowledge and understanding of the principles, objectives and procedures of the PMDS and their satisfaction with the system. The study also focused on capacitating the respondents in this regard. Further research is required to gather the respondents’ qualitative insights on the strengths and weaknesses of the system, that is, the respondents’ in-depth views regarding the value, purpose, efficiency and effectiveness of the system. 

### 5.2. Ethical Considerations 

The researcher upheld ethical standards by conducting the research in accordance with standard ethical guidelines. Permission to conduct the study was sought from the Chief Executive Officer of Brits District Hospital. Approval to conduct the study was sought from the Sefako Makgatho Health Sciences University Research and Ethics Committee. The Sefako Makgatho Health Sciences University Research and Ethics Committee reference number is SMUREC/H/273/2019: PG. Written consent to participate in the study was obtained from the respondents. The confidentiality of the respondents was ensured by not publishing any of their names in the study because they may have been concerned that what they wrote in the questionnaire might expose them to prejudice in the workplace. The research procedure was explained, and the respondents were given a chance to read through the covering letter before verbal and written consent was requested from them. The respondents were informed about their right to withdraw from the study at any time. The respondents were not required to give their names in the questionnaire.

## Figures and Tables

**Figure 1 ijerph-19-14461-f001:**
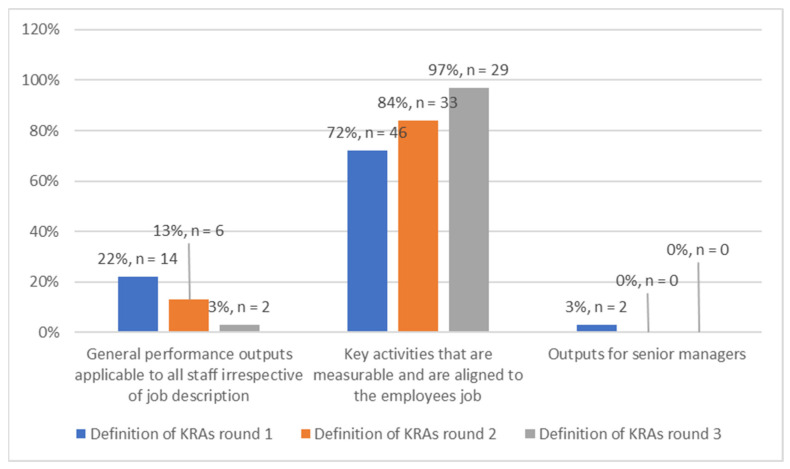
Definition of Key Responsibility Areas (*n* = 135).

**Figure 2 ijerph-19-14461-f002:**
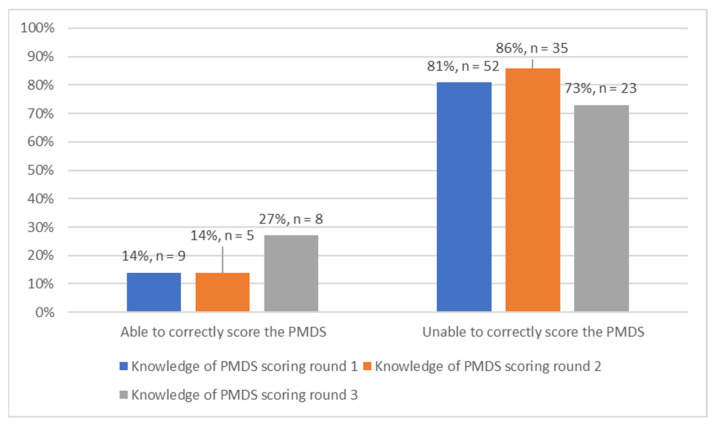
Knowledge of PMDS scoring (*n* = 135).

**Figure 3 ijerph-19-14461-f003:**
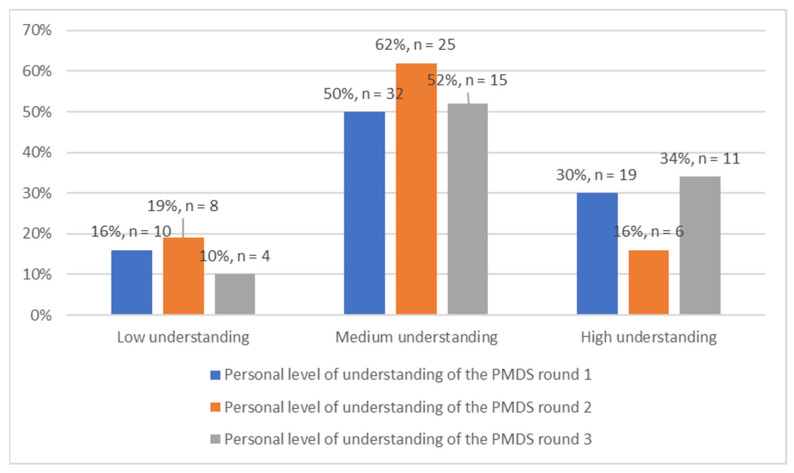
Personal level of understanding of the PMDS (*n* = 135).

**Figure 4 ijerph-19-14461-f004:**
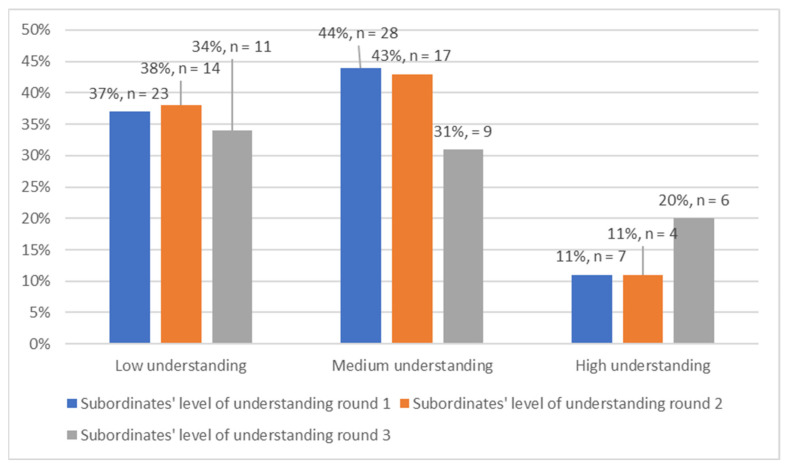
Subordinates’ level of understanding (*n* = 135).

**Figure 5 ijerph-19-14461-f005:**
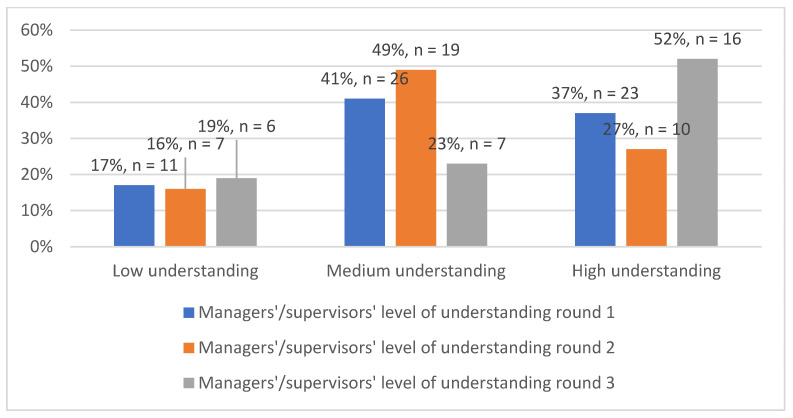
Managers’/supervisors’ level of understanding of the PMDS (*n* = 135).

**Table 1 ijerph-19-14461-t001:** Superiors’ understanding of the PMDS and your involvement in a PMDS dispute (round three).

Variable	Obs	Mean	Std. Err.	Std. Dev.	[95% Conf. Interval]	
Unders Sub	31	2.193548	0.1760948	0.9804541	1.833915	2.553182
Dispute	31	1.903226	0.0539781	0.3005372	1.792988	2.013464
Combined	62	2.048387	0.0932049	0.7338964	1.862021	2.234762
diff		0.2903226	0.184182		−0.0833659	0.664011
diff: = mean (Unders You) − mean (Dispute) t = 1.5763						
Ho: diff = 0 Satterthwaite’s degrees of freedom = 34.3363						
Ha: diff < 0 Ha: diff! = 0 Ha: diff > 0						
Pr (T < t) = 0.9381 Pr (|T| > |t|) = 0.1238 Pr (T > t) = 0.0619						

## Data Availability

The data presented in this study are available in Appendix A.

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
