# Peer review of "Knowledge, Understanding and Satisfaction with the Implementation of the Performance Management System at a District Hospital in the Madibeng Subdistrict, South Africa"

_ijerph, 2022, doi:10.3390/ijerph192114461_

Round 1
Reviewer 1 Report
Dear authors, after reading your work I have the following recommendations that maybe could help you to improve this work:
General comments:
The topic under analysis is interesting and relevant for the journal. The objective of this study was to determine the perceived knowledge, understanding and satisfaction levels of employees at a Hospital regarding the Performance Management and Development System.
The paper is well written, the subject address is current and pertinent, and more research should be done on this topic in the literature. As such, the paper could have a place in International Journal of Environmental Research and Public Health editorial line.
Specific comments:
The title is specific and relevant but very long.
Is clear the reading of the text (textual coherence and cohesion).
The abstract it is well written, and we can understand the question addressed in a broad context, the purpose of the study, the main method, and the main findings.
The introduction section must explain in a convincing manner the relevance of the paper, especially regarding its academic and practical implications. Nowadays, there is a great number of performance management and development system. Why these frameworks are not enough? Why study Brits District Hospital?
The introduction should include the theme, the motivation for the choice of the problem, the objectives, the research question, the methodology to be followed and the work structure. Thus, a contextual discussion of the work must be done, justifying the theoretical, social and practical importance of the research problem. Citations should be avoided as much as possible.
The bibliographic references aren’t current and relevant. I suggest the authors to include a section with recent and relevant references about the need and importance to implement performance management systems in public health care institutions.
In the methodology the authors must situate the study on the time. It’s important describe the methodology with sufficient detail with dates to allow others to replicate and build on published results. It is important explain when the study was carried out.
I suggest the authors to describe in detail in the methodology section the questions included in the questionnaires.
The results are essentially descriptive, why the authors don’t define and statistically test a range of hypothesis?
I suggest complete the discussion section with recent references.
The conclusions adequately tie together the other elements of the paper however must emphasis very well the contributions of the study.
I appreciate that the authors identify and suggest future research in this topic.
I hope that my comments and suggestions can help to improve the paper.
Thank you for the opportunity to read your article.
I wish all the best to the authors!
Reviewer 2 Report
The paper undertakes the subject of PMDS assessment by employees. The subject is original and scientifically interesting, nonetheless there are issues which must be improved.
- The introduction does not clearly state the aim of the paper. It is unclear what the Authors would like to achieve.
- Introduction lacks explanation of what a PMDS is (it is presented in the abstract) and it presents it very subjectively – only negative aspects are emphasized. Therefore, PMDS should be directly defined and presented more objectively.
- In introduction there should be, at least, limited discussion of Brits District Hospital’s characteristics, which could impact the quantitative research.
- There is no discussion of previous or similar empirical research conducted in this area. In reviewer’s opinion, the introduction should be divided into two separate parts – introduction and literature review – with each of them developed more carefully.
- The ethical considerations should be moved to the end of the article.
- In the empirical part it is not clear what was the time break between the rounds of data collection. The time span could have an impact on respondents answers. [I see that in Limitations part it states that the time span was 3 months, but it should be written and explained in the methodological part.)
- The language should be improved. It is inappropriate to use contractions (e.g. don't, aren't) in academic research.
Reviewer 3 Report
The topic is interesting. However, the following issues should be solved to improve the manuscript quality:
1. Typos, e.g. "The employees where..." (Line 22-23); missing appropriate punctuations, e.g. Lines 113, 116, etc.
2. The figures are not numerically ordered and referenced.
3. The results shown in Section 3 should be accompanied with sound explanations.
4. Section 4 should provide the at least a new finding compared to other published studies to further emphasize the key contribution from this research.
Round 2
Reviewer 1 Report
The author made all the changes suggested in the initial revision of the paper in a satisfactory manner, significantly improving the previous version of the paper.
Reviewer 2 Report
Proofreading of the final text is required. Also, some minor style changes such as beginning of a section with a citation [17].
Reviewer 3 Report
There is a significant improvement with the revised version. However, please double check for typos and grammar/ punctuation usage.